# Breast Milk for Term and Preterm Infants—Own Mother’s Milk or Donor Milk?

**DOI:** 10.3390/nu13020424

**Published:** 2021-01-28

**Authors:** Réka A. Vass, Gabriella Kiss, Edward F. Bell, Robert D. Roghair, Attila Miseta, József Bódis, Simone Funke, Tibor Ertl

**Affiliations:** 1Department of Obstetrics and Gynecology, University of Pécs Medical School, 7624 Pécs, Hungary; bodis.jozsef@pte.hu (J.B.); funke.simone@pte.hu (S.F.); tibor.ertl@aok.pte.hu (T.E.); 2MTA-PTE Human Reproduction Scientific Research Group, University of Pécs, 7624 Pécs, Hungary; 3Department of Laboratory Medicine, University of Pécs Medical School, 7624 Pécs, Hungary; kiss.gabriella2@pte.hu (G.K.); miseta.attila@pte.hu (A.M.); 4Stead Family Department of Pediatrics, University of Iowa, Iowa City, IA 52242, USA; edward-bell@uiowa.edu (E.F.B.); robert-roghair@uiowa.edu (R.D.R.)

**Keywords:** preterm newborns, nutrition, breast milk, infant formula, FSH, LH, PRL, pituitary hormones

## Abstract

Hormones are important biological regulators, controlling development and physiological processes throughout life. We investigated pituitary hormones such as follicle-stimulating hormone (FSH), luteinizing hormone (LH), prolactin (PRL) and total protein levels during the first 6 months of lactation. Breast milk samples were collected every fourth week of lactation from mothers who gave birth to preterm (*n* = 14) or term (*n* = 16) infants. Donor milk is suggested when own mother’s milk is not available; therefore, we collected breast milk samples before and after Holder pasteurization (HoP) from the Breast Milk Collection Center of Pécs, Hungary. Three infant formulas prepared in the Neonatal Intensive Care Unit of the University of Pécs were tested at three different time points. Our aim was to examine the hormone content of own mother’s milk and donor milk. There were no significant changes over time in the concentrations of any hormone. Preterm milk had higher PRL (28.2 ± 2.5 vs. 19.3 ± 2.3 ng/mL) and LH (36.3 ± 8.8 vs. 15.9 ± 4.1 mIU/L) concentrations than term milk during the first 6 months of lactation. Total protein and FSH concentrations did not differ between preterm and term breast milk. Holder pasteurization decreased the PRL concentration (30.4 ± 1.8 vs. 14.4 ± 0.6 ng/mL) and did not affect gonadotropin levels of donor milk. Infant formulas have higher total protein content than breast milk but do not contain detectable levels of pituitary hormones. Differences were detected in the content of pituitary hormones produced for preterm and term infants. Divergence between feeding options offers opportunities for improvement of nutritional guidelines for both hospital and home feeding practices.

## 1. Introduction

Human milk is considered the best source of nutrition for infants, in part because it contains hormones, growth factors, cytokines, and immunoglobulins [1,2]. Hormones are important regulators of physiological functions and development from the beginning of life [3]. During intrauterine life, maternal hormones expose the embryologic development through transplacental circulation, and it is also known that amniotic fluid is a transport milieu of hormones [4,5]. After birth, breast milk is the exclusive source of maternal hormones, exerting their effects via intestinal absorption or locally in gastrointestinal tract [6,7].

In the hospital and at home, two options are available to provide breast milk for a child, its own mother’s milk or, in case this is not available, providing donor milk. Donor milk, to ensure its microbiological safety, is Holder pasteurized, which influences its composition [8,9,10]. It is also known that milk of mothers who gave birth to preterm infants differs from breast milk produced for term infants [11]. Regulatory hormones, like the pituitary hormones, are present in breast milk [9], but no data are available regarding the monthly concentration changes of prolactin (PRL), follicle-stimulating hormone (FSH), and luteinizing hormone (LH) during the first 6 months of lactation. PRL is mainly synthesized by lactotrophs, while FSH and LH are produced by the gonadotroph cells in the anterior lobe of the pituitary gland. PRL is a neuropeptide and a neurotransmitter with a reproductive role, influencing luteal function, reproductive behavior, homeostasis, osmotic balance, and angiogenesis throughout life [12].

In fetal life, marked changes in the hypothalamic-pituitary-gonadal (HPG) axis hormone activity occur mainly in the second trimester of pregnancy, with the secretion of pituitary FSH and LH increasing already from week 10 of gestation. Minipuberty, the transient reactivation of the HPG-axis, occurs at the beginning of postnatal life. The gonadotropins, FSH and LH, peak throughout the first 3 months of age in both sexes, followed by a significant suppression until puberty. This early period of FSH and LH production is considered a critical window of sex-dependent genital development. Females have higher FSH versus LH plasma levels compared to male fetuses and infants throughout this period. In female infants, early pituitary glycoprotein exposure through impaired negative feedback can lead to ovarian torsion, while in males, later hypogonadism may develop due to disruption of minipuberty [13].

We aimed to examine the presence of PRL, FSH, and LH in preterm and term milk during the first 6 months of lactation, and to investigate the effects of Holder pasteurization (HoP) on the hormone content in donor breast milk samples. When human milk is not available to feed a child or if the parents prefer not to use donor milk, infant formula is used. Therefore, we completed our investigation by measuring the presence of pituitary hormones in the third available feeding option, infant formula. The purpose of our project was to provide information about an important group of hormones with numerous regulatory effects in own mother’s milk, donor milk, and infant formula.

## 2. Methods

This study was approved by the Regional and Local Research Ethics Committee of the University of Pécs, Pécs, Hungary (PTE KK 7072-2018). Written informed consent was obtained from all participants.

Mothers who gave birth to term (*n* = 16) and preterm (*n* = 14) infants were enrolled at the Department of Obstetrics and Gynecology, University of Pécs. Breast milk collection commenced at 4 weeks postpartum and continued every 4 weeks until 6 months postpartum. Participants pumped the entire breast expression into a sterile bottle between 1 pm and 3 pm, and 5 mL was poured into polypropylene tubes.

In the second part of the study, we recruited 40 registered and approved donor mothers from the Milk Bank of the Unified Health Institution at Pécs, Hungary. They donated freshly pumped breast milk. At the morning pumping, fresh samples were taken before pooling at the Milk Bank. We collected samples in six different time points. After pooling, the milk samples were Holder pasteurized based on the protocol of the Unified Health Institution. For laboratory analysis, three samples were taken from the pooled pasteurized breast milk. All samples were stored at −80 °C in sterile polypropylene tubes until measurements were completed.

Holder pasteurization was performed in the Milk Bank of the Unified Health Institute, breast milk was pasteurized in a professional water bath system for 30 min at 62.5 °C. Each sample was sonicated to disrupt the milk fat globule membranes and centrifuged at 15,000× *g* for 15 min; then the skim milk was transferred to tubes for the analyses, as previously described [9,10].

For comparison, three infant formulas prepared in the Neonatal Intensive Care Unit of the University of Pécs were tested at three different time points: Nutricia Milumil Pepti Pronutra (Danone, Paris, France), Beba Optipro Hypoallergenic (HA) Start (Nestlé, Vevey, Vaud, Switzerland), and Beba Optipro HA Pre (Nestlé, Vevey, Vaud, Switzerland).

For PRL detection a 2-step immunoassay was applied. First, 66 µL of monoclonal anti-prolactin coated microparticles was added to 10 µL of breast milk sample. After washing, 59 µL of an anti-prolactin monoclonal acridinium labeled conjugate was added, and a sandwich complex was formed. Pre-Trigger Solution hydrogen peroxide and Trigger Solution sodium hydroxide were added to the mixture. This resulted in a chemiluminescent reaction that was detected and measured as relative light units by the ARCHITECT i optical system (Abbott Laboratories, Abbott Park, IL, USA). PRL detection range was 0.6–200 ng/mL.

For LH detection, a similar immunoassay was used. First, 10 µL of sample was mixed with 66 µL anti-beta LH-coated paramagnetic microparticles. After washing steps, 59 µL of anti-alfa acridinium-labeled conjugate was added. The chemiluminescent reaction was detected as relative light units, measured by the ARCHITECT i optical system. The applied 2-step immunoassay’s measuring range for LH was 0.09–250 mIU/mL.

To measure FSH, at first 40 µL of sample was mixed with anti-beta FSH coated microparticles. After washing steps, anti-alfa FSH acridinium labeled conjugate was added. Pre-Trigger and Trigger solutions were added, and the resulting chemiluminescent reaction was detected and measured with the ARCHITECT i optical system. The detection range was between 0.05 and 150 mIU/mL.

For the measurements we applied the ARCHITECT i system and followed the manufacturer’s instructions. All measurements were performed with the fully automatized Cobas e 411 analyzer system (Roche Diagnostics, Rotkreuz, Switzerland). With this system, application of a voltage to the electrode induces chemiluminescent emission, which is measured by a photomultiplier. Results were determined via a calibration curve, which is instrument specific and generated by 2-point calibration, and a master curve provided based on the reagent barcode. Quality controls were tested in parallel with the breast milk samples. In case of applied monoclonal antibodies, the following cross-reactivity values were detected: LH, thyroid stimulating hormone (TSH), Human chorionic gonadotropin (hCG), human growth hormones (hGH), and human placental lactogen (hPL) < 0.1%. The lower detection limits were for PRL 1.00 µIU/mL (0.047 ng/mL), LH 0.05 mIU/mL, and FSH 0.05 mIU/mL.

Total protein was measured in samples using the biuret method, a colorimetric technique suitable for the determination of total protein by spectrophotometry (at 540–560 nm).

We applied GraphPad (La Jolla, CA, USA) for statistical analyses, and Saphiro—Wilks tests were used to test the normality of data. Statgraphics Centurion XVII, v17.0.16 (Statpoint Technologies, Inc., Warrenton, VA, USA), and R v3.3.2 (R-project, Vienna, Austria http://www.r-project.org) software were used to perform the data analyses. Data were analyzed by using paired t-test or analysis of variance test (ANOVA). Infant gender was analyzed by Fisher’s exact test. Repeated measures one-way ANOVA test (with post-hoc Dunnet test) was used to compare the effect of HoP on the concentration of these compounds. Concentration values below the lower limit of quantification (LLOQ) were assigned values equal to the LLOQ divided by the square root of 2 [4,5]. Differences were considered statistically significant when *p* values were <0.05. The study was powered to detect moderate effect sizes (Cohen’s d = 0.6). Results are shown as mean ± standard error of the mean (SEM) or median values with interquartile ranges.

## 3. Results

### 3.1. Maternal Demographics

Maternal age, body-mass index (BMI) and infant sex did not show differences in comparisons between preterm and term groups. Mothers who donated their milk samples to the milk bank were more likely to have given birth to male infants. In our study, all recruited mothers were Caucasians, and none had diabetes before or during their pregnancy or throughout the study, and none followed any special diet. As expected, gestational age of the preterm group was significantly lower (*p* < 0.001) compared to the term and donor groups (Table 1). Mothers who gave birth to term and preterm infants gave breast milk samples in every month after delivery, whereas the age at donation for the donor mothers was about 5 months (145.2 ± 12.9 days).

### 3.2. Preterm and Term Breast Milk at the Time of Recommended Exclusive Breastfeeding

Total protein level was similar in breast milk produced for term and preterm infants. Preterm breast milk had significantly higher PRL, FSH, and LH content than did term milk during the first 6 months of lactation (Table 2).

### 3.3. Effects of Holder Pasteurization

Holder pasteurization did not affect LH, FSH, or total protein content of breast milk. The mean PRL after HoP was 47.4% of the pre-pasteurization value in the donor milk samples (Table 3).

### 3.4. FSH:LH Ratio in Breast Milk

Preterm and term breast have similar FSH:LH ratios throughout the first 6 months of lactation, and no differences were detected between the two groups (*p* = 0.789). Donor mothers gave birth to term infants; although the time points of collection were different, the FSH:LH ratio in unpasteurized donor milk was similar to other two groups (*p* = 0.127 vs. preterm and *p* = 0.214 vs. term). Pasteurized donor milk had a significantly lower FSH:LH ratio compared to the preterm (*p* = 0.003), term (*p* = 0.005), and unpasteurized donor milk (*p* = 0.025) (Figure 1).

### 3.5. Investigation of Infant Formula

Milumil Pepti Pronutra, Beba Optipro HA Start, and Beba Optipro HA Pre infant formulas contained the pituitary hormones below the lower limit of detection. When comparing different formulas, no difference in hormone levels was detected. Analyzing pooled results, formulas had an average 15.93 ± 1.06 g/L total protein level, which was significantly higher compared to term (*p* < 0.0001) and preterm (*p* < 0.0001) breast milk throughout the first 6 months of lactation.

### 3.6. Exploratory Analysis of Confounding Variables

Infant gender, maternal age, maternal BMI, and mode of delivery had no impact on pituitary hormone or protein content of human milk when comparing preterm and term breast milk samples monthly or pooled.

## 4. Discussion

In the present study, we detected the presence of pituitary hormones FSH, LH, and PRL throughout the first 6 months of lactation in breast milk produced for term or preterm infants. These hormones were measured in donor breast milk, and commercial infant formulas as well, to provide previously unavailable information about the FSH, LH and PRL concentration of feeding options during postnatal life. Preterm breast milk contained a significantly higher amount of PRL, while FSH and LH concentrations were measured in the same amount in preterm and term breast milk. Moreover, Holder pasteurization applied to ensure microbiological safety of donor milk had a significant impact on PRL concentration. Comparing different feeding options, formula contained undetectable amounts of pituitary hormones, and donor milk contained lower PRL than own mother’s breast milk. It is important to note that the sensitivity of laboratory methods are constantly developing; therefore, measurements of the bioactive compounds in breast milk are necessary to provide detailed information about the postnatal influences impacting the newborn and also to help improve donor milk supplements and develop better infant formulas. Our results, due to the detected differences between the hormonal intake from different feeding sources, as it was earlier concluded [14], strengthen the possibility that hormone supplementation might be a building block of nutritional strategies in neonatal care. An elegant study highlighted in the 1990s that breastfed preterm infants achieved better results on later IQ tests [15]. The reasons for this are not known, but they may include the higher hormonal content of human milk.

Nutrients, including immunoglobulins, vitamins, growth factors, chemokines, cytokines, oligosaccharides, and immune cells, have been demonstrated to transfer from the mother to the neonate through breast milk [1,16,17]. The increasing knowledge of the important relationship of breastfeeding to infant outcomes highlights the value of breastfeeding in postnatal human development.

All hormones of interest were detected in breast milk samples. Previously, our research group investigated and reported the presence of gonadotropins in breast milk [9], and PRL is a known component of human milk [18]. Concentration of PRL was detected two times higher in a study investigating its presence with radioimmunoassay on the fourth day of lactation [19]. Previous investigation showed no daily fluctuation in concentration of PRL in breast milk during the first month of breastfeeding [19]. In females, serum prolactin levels during the third trimester of pregnancy range from 10 to 210 ng/mL. PRL receptors are expressed on small intestine epithelial cells in rabbits and rats throughout neonatal period and maternal radiolabeled PRL was detected in pup’s circulation after suckling, moreover, PRL appeared biologically active [20,21,22]. PRL is synthesized in the placenta and transferred via amniotic fluid; after birth, PRL is transferred to the infant through breast milk. Formula-fed preterm neonates display elevated fasting serum PRL immunoreactivity and reduced PRL bioactivity compared to their human milk-fed counterparts [23]. Ovarian dysgenesis is common in female mice with genetically reduced serum PRL, and altered sexual maturity may develop in neonates ingesting PRL-poor milk [24]. In adult male mice, the weights of the testes and adrenals were significantly reduced; in female mice, opening of the vagina was delayed when neonates were treated with anti-PRL sera [24]. The expression of PRL receptors in neonatal thymus and spleen was modified by milk intake rate [25]. Previous results indicated that amniotic and human milk PRL excretion may contribute to the regulation of the sodium and chloride plasma concentrations in the newborn [26,27]. It was also concluded that PRL may be involved in control of the volume and composition of body fluids in the neonate [28]. A notable difference was measured between the PRL concentration of breast milk produced for term and preterm infants. Higher breast milk PRL concentration may have resulted from physiological changes leading to preterm labor. Previous investigators found that maternal plasma PRL level was significantly higher in mothers delivering preterm than at term [29].

The gonadotropic hormones, FSH and LH, control reproductive processes and also have impact on neurodevelopment, and there are important consequences of deficiencies of these hormones during both prenatal and neonatal development [30,31,32]. LH levels exceed FSH concentration in male fetuses [33]. In the background of gender difference, the regulatory negative feedback is expected in males through higher levels of fetal testicular hormones [34,35]. In females, FSH levels are higher than in males, and a peak was observed between the first week and 3 months of age. Afterwards, the FSH concentrations gradually decrease within 4 months to the prepubertal range, while FSH level remains elevated in females until 3–4 years of age [36,37]. Elevated LH levels were detected in the preterm population [38,39]. Based on our findings, breast milk produced for preterm infants has higher LH concentration compared to term milk. Previous studies have investigated the role of infant nutrition on plasma FSH and LH levels. Fang and colleagues showed higher FSH and lower LH levels in boys receiving breast milk compared to cow or soy formula [40]. Breast milk is an important carrier of pituitary hormones compared to infant formulas, which lack pituitary hormones. Follow-up studies examining adult reproductive outcomes might show possible later effects. The FSH:LH ratios we found in preterm and term breast milk were similar to those previously reported in maternal and donor breast milk [9]. Although the Holder pasteurization did not significantly affect the levels of FSH and LH in breast milk, the procedure lowered the FSH:LH ratio compared to that found in preterm (*p* = 0.003), term (*p* = 0.005), and unpasteurized donor milk (*p* = 0.025) (Figure 1).

Gastric digestion of proteins and intestinal digestion of other components are attenuated during the first 3 postnatal months; in the infant’s gastrointestinal tract, the barrier functions and the cell-cell connections are immature. Therefore, pituitary hormones such as PRL, FSH and LH can exert their effects in the body through absorption from the gastrointestinal tract [6,7]. Previous investigators found that PRL [12] and LH can cross the blood-brain barrier [41,42], are present in cerebrospinal fluid [43,44], and can be produced by extra-hypothalamic neurons [12].

As found by previous investigators [38,39], we found elevated levels of LH in the breast milk of mothers who delivered prematurely compared to term milk. Movsas and coworkers observed that elevated plasma LH is associated with the development of retinopathy of prematurity in female preterm infants but not male infants [39]. In another investigation, increased secretion of FSH was correlated with the prevalence of infantile hemangioma [45]. These results highlight that the sensitive balance of hormone levels may be crucial during this critical window of postnatal adaptation. Previous results indicate that the excretion of PRL into amniotic and human milk may be important for the regulation of the plasma sodium and chloride concentrations of the newborn [25,26].

If own mother’s milk is not available in the neonatal intensive care unit, donor milk is the recommended choice for preterm infants. Before feeding it to infants, donor milk is submitted to HoP in human milk banks to ensure microbiological safety. The impact of HoP on the concentration of various hormones differs. Some hormones such as insulin and leptin decrease, while others are not influenced by the procedure [10]. Our finding that protein content of donor milk did not change significantly due to HoP agreed with several previous studies [46,47], while another study showed significant decrease in protein concentration following HoP [48]. Different studies applying different methods or recruiting mothers in different continents described varied effects of HoP [49]. Our data strengthen the idea that hormone content is influenced by HoP in unique ways.

Our study has limitations. The chosen technique is the preferred method to measure biologically active hormone levels in clinical diagnostic laboratories. Removal of the fat layer during the sample preparation may reduce detectability, but this method, as in our previous projects [9,10] was validated for nonaqueous samples.

Although nutritional therapeutic options for preterm infants are constantly improving, these infants are still often faced with the consequences of postnatal growth restriction. Hormones have numerous physiological impacts on development, so it should be considered whether supplementation of infant feeding regimens with selected hormones might offer benefits to infant nutrition and health. Ng and coworkers recently published a study investigating the effect of early thyroxine supplementation and found that it improves neurodevelopment at age 3–4 years in infants born below 28 weeks’ gestation [50]. This observation highlights the importance of studies focusing on early supplementation, especially in donor milk-fed or formula-fed preterm infants.

The remarkable differences in composition among preterm and term infant’s own mother’s milk, pasteurized donor milk, and infant formulas suggest opportunities for improvement of nutritional guidelines in clinical practice.

## 5. Conclusions

The present findings expand our knowledge about the concentrations of different pituitary hormones transferred to the infant through breastfeeding. Lack of the third trimester’s hormonal impact through placental transfer resulting from preterm birth places the preterm infant into an altered developmental milieu after birth. Preterm infants suffer from a lack of hormonal impact from the mother’s body. Because the hormones normally transferred from the mother during the third trimester are lost, nutrition and the use of a biologically active fluid like breast milk may be critically important as a source of hormones, in preference to infant formula.

Our findings demonstrate, for the first time, that three pituitary protein hormones, PRL, FSH, and LH are present in preterm and term breast milk throughout the first 6 months of lactation and that Holder pasteurization technique influences the concentrations of PRL, FSH, and LH in hormone-specific ways. Clinical investigations should examine the implication of feeding options on the hormonal status of newborns.

## Figures and Tables

**Figure 1 nutrients-13-00424-f001:**
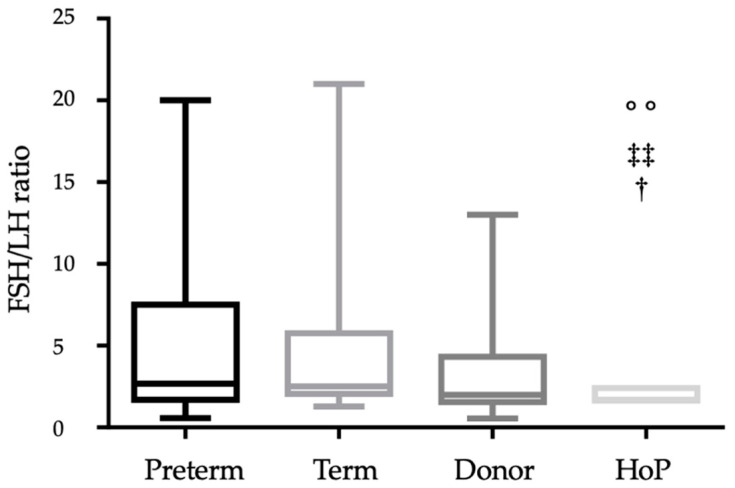
FSH:LH ratio in preterm, term, donor, and pasteurized donor milk. ^°°^
*p* < 0.01 versus preterm milk; ^‡‡^
*p* < 0.01 versus term milk; ^†^
*p* < 0.05 versus donor milk.

**Table 1 nutrients-13-00424-t001:** Clinical data of enrolled patients.

	Preterm Maternal	Term Maternal	Donor
Maternal age (years)	31.7 ± 1.1	32.1 ± 2.7	32.4 ± 0.5
Gestational age (weeks)	31.4 ± 2.1	39.6 ± 0.5	39.5 ± 0.2
Maternal BMI	27.8 ± 0.3	26.9 ± 0.7	26.3 ± 1.8
Gender of newborn			
Female	7	9	17
Male	7	7	24
Mode of delivery			
Vaginal	4	11	26
C-section	10	5	15

**Table 2 nutrients-13-00424-t002:** Total protein, prolactin (PRl), luteinizing hormone (LH), and follicle-stimulating hormone (FSH) concentration in term and preterm milk during the first 6 months of lactation.

Analyte	Term (*n* = 96)	Preterm (*n* = 84)	*p*-Value
Total protein g/L	9.54 ± 0.3	10.13 ± 0.3	0.207
PRL ng/mL	19.31 ± 2.2	28.24 ± 2.5	0.012
LH mU/L	15.86 ± 4.1	36.27 ± 8.8	0.041
FSH mU/L	122.92 ± 8.8	178.94 ± 14.3	0.001

**Table 3 nutrients-13-00424-t003:** Impact of Holder pasteurization on the concentrations of FSH, LH, and PRL in donor milk (*n* = 41).

Analyte	Raw	HoP	*p*-Value
Total protein g/L	9.65 ± 0.1	9.93 ± 0.2	0.217
PRL ng/mL	30.37 ± 1.7	14.39 ± 0.6	0.0001
LH mU/L	67.01 ± 11.2	55.54 ± 6.1	0.339
FSH mU/L	138.69 ± 7.4	133.39 ± 4.7	0.621

## Data Availability

The data presented in this study are available on request from the corresponding author.

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
