# Peer review of "Breast Milk for Term and Preterm Infants—Own Mother’s Milk or Donor Milk?"

_nutrients, 2021, doi:10.3390/nu13020424_

Round 1
Reviewer 1 Report
Interesting and well presented paper.
One statistical issue: the authors test normality of distributions but they did not specify with kind of analysis they performed for mean comparisons. although intuitively understandable, it should be described.
No other major issues.
Author Response
Answers for Reviewer 1:
1.) Interesting and well presented paper.
Thank you.
2.) One statistical issue: the authors test normality of distributions but they did not specify with kind of analysis they performed for mean comparisons. Although intuitively understandable, it should be described.
We thank you for this comment. We have added the following to the Methods section: “Data were analyzed by using paired t-test or ANOVA. Infant gender was analyzed by Fisher’s exact test. Repeated measures one-way ANOVA test (with post-hoc Dunnet test) was used to compare the effect of HoP on the concentration of these compounds.” (lines 128-130)
Reviewer 2 Report
Review Nutrients 1090537
Thank you for the opportunity to review this paper. The importance of breastmilk for optimal infant nutrition cannot be understated, and we have a long way to go to understand the full complexity of the components of breastmilk and the role of these components in the development of the infant, both term and preterm. The results of this study provide a few more pieces in this puzzle.
I have some suggestions that I think will improve this paper.
Please quote reference(s) to support, “Hormones are important regulators of physiological functions and development from the beginning of life.”
And also: “During intrauterine life, maternal hormones expose the embryologic development through transplacental circulation, and it is also known that amniotic fluid is a transport milieu of hormones.”
The transfer of bioactive components of milk to the infant via intestinal absorption should be mentioned in the Introduction, rather than waiting until the Discussion. That will provide the reader with a reason for measuring the hormones in the breastmilk.
Line 38
Instead of “breastfeed a child
Use, “provide breastmilk for a child”
Have any differences in development been observed between infants fed breastmilk (Mothers’ own milk or donor milk) and infant formula?
Methods for analysis of PRL, LH and FSH should be referenced. If they were developed by the researchers more detail is required, particularly sources of reagents, accuracy and precision.
Were there any trends in hormone concentrations from 1 to 6 months postpartum for either pre-term or term milk, given the statement in the Discussion, “In females, FSH levels are higher than in males, and a peak was observed between the first week and 3 months of age. Afterwards, the FSH concentrations gradually decrease within 4 months to the prepubertal range, while FSH level remains elevated in females until 3–4 years of age [29,30].”
Table 2
Are the given concentrations the mean ± SEM for all samples within each group?
A significant difference is shown in PRL concentration between term and preterm infants, but this is not discussed.
Please explain the statement, “Pasteurized donor milk had a significantly lower FSH:LH ratio compared to the preterm (p=0.003), term (p=0.005), and unpasteurized donor milk (p=0.025) (Fig. 1).
Using the data from Table 3:
For raw milk I calculate the FSH:LH ratio as 138.69/67.01 = 2.06
For HoP milk I calculate the FSH:LH ratio as 133.39/55.54 = 2.40
which is not lower.
Discussion
Apart from the more minor comments, above, I think the Discussion needs a major rewrite to provide clarity for the reader.
Reference 9 reviews maternal diet and breast milk composition with respect to total energy, total protein, fat, fatty acids, carbohydrates, vitamins and minerals, but makes no reference to, “immunoglobulins, growth factors, cytokines, and immune cells”
The same applies to reference 10.
Lines 192-193
“Prolactin is the major driver of development during pregnancy via regulation of ovarian progesterone production (in many species) and direct effects on mammary epithelial cells (in all species)”
is a direct quote from the abstract of reference 13. However, it is referring to mammary gland development and not fetal or newborn development.
In the Discussion it would be preferable to highlight the implications of the results of the current study rather than the in depth description of the many findings from the literature with respect to actions of the hormones of interest on fetal and neonatal development.
Author Response
Answers for Reviewer 2:
1.) Please quote reference(s) to support, “Hormones are important regulators of physiological functions and development from the beginning of life.”
Thank you for the suggestion. We added this citation (Ref. #3) Forest, M.G.; De Peretti, E.; Bertrand, J. Hypothalamic-pituitary-gonadal relationships in man from birth to puberty. Clin Endocrinol (Oxf) 1976, 5, 551-569. doi: 10.1111/j.1365-2265.1976.tb01985.x.
2.) And also: “During intrauterine life, maternal hormones expose the embryologic development through transplacental circulation, and it is also known that amniotic fluid is a transport milieu of hormones.”
We cited these reports (Refs. #4, 5) Fowden, A.L.; Forhead, A.J.; Coan, P.M.; Burton, G.J. The placenta and intrauterine programming. J Neuroendocrinol 2008, 20, 439-450. doi: 10.1111/j.1365-2826.2008.01663.x.
Dawood, M.Y. Hormones in amniotic fluid. Am J Obstet Gynecol 1977, 128, 576-583. doi: 10.1016/0002-9378(77)90046-1.
3.) The transfer of bioactive components of milk to the infant via intestinal absorption should be mentioned in the Introduction, rather than waiting until the Discussion. That will provide the reader with a reason for measuring the hormones in the breastmilk.
We are grateful for this observation. The following sentence was added to the Introduction: “After birth, breast milk is the exclusive source of maternal hormones, exerting their effects via intestinal absorption or locally in the gastrointestinal tract (Weaver et al, 1984; Oguchi et al, 1997).” (lines 39-40).
4.) Line 38 - Instead of “breastfeed a child Use, “provide breastmilk for a child”
We thank and accept this suggestion and changed the language accordingly.
5.) Have any differences in development been observed between infants fed breastmilk (Mothers’ own milk or donor milk) and infant formula?
In previous elegant study, children who had consumed their own mother's breast milk postnatally had a significantly higher IQ at 7 1/2-8 years compared to those who received no breast milk (Lucas et al, 1992). Our study protocol will be examining the recruited children at the age of 1 year, about weight, height, and body composition. Data collection is continuing.
Lucas, A.; Morley, R.; Cole, T.J.; Lister, G.; Leeson-Payne, C. Breast milk and subsequent intelligence quotient in children born preterm. Lancet 1992, 339, 261-264. doi: 10.1016/0140-6736(92)91329-7.
6.) Methods for analysis of PRL, LH and FSH should be referenced. If they were developed by the researchers more detail is required, particularly sources of reagents, accuracy and precision.
The following sentences were added to the Methods section:
“For the measurements we applied the ARCHITECT i system (Abbott Laboratories, Abbott Park, Illinois, United States) and followed the manufacturer’s instructions.” (line 125-126)
“During analysis of skim milk we followed the manufacturer’s instructions.” (line 98-99) and “In case of applied monoclonal antibodies, the following cross-reactivity values were detected: LH, TSH, hCG, hGH and hPL <0.1 %.” (lines 134-136).
Skim milk was used for the measurements, since the composition of skim milk is closer to plasma. After consulting the company and based on previous studies (Steinbrekera et al, 2020; Vass et al, 2020a; Vass et al, 2020b) we followed the official instructions.
We noted this information in the Discussion as a limitation of our investigation: (lines 264-269) “Our study has limitations. The chosen technique is the preferred method to measure biologically active hormone levels in clinical diagnostic laboratories. Removal of the fat layer during the sample preparation may reduce detectability, but this method, as in our previous projects (Vass et al, 2020; Vass et al, 2020) was validated for nonaqueous samples.”
7.) Were there any trends in hormone concentrations from 1 to 6 months postpartum for either pre-term or term milk, given the statement in the Discussion, “In females, FSH levels are higher than in males, and a peak was observed between the first week and 3 months of age. Afterwards, the FSH concentrations gradually decrease within 4 months to the prepubertal range, while FSH level remains elevated in females until 3–4 years of age [29,30].”
During our measurements, monthly concentrations of FSH and LH did not change, but we were concerned that the fluctuations of FSH and LH are essential and interesting details that should be noted to the readers.
8.) Table 2 Are the given concentrations the mean ± SEM for all samples within each group? A significant difference is shown in PRL concentration between term and preterm infants, but this is not discussed.
We gave the mean ± SEM values, and we added the following part to the Discussion section (lines 346-350): “A notable difference was measured between the PRL concentration of breast milk produced for term and preterm infants. Higher breast milk PRL concentration may have resulted due to physiological changes leading to preterm labor. Previous investigators found that maternal plasma PRL level was significantly higher in mothers delivering preterm than at term (Mazor et al, 1996).
Mazor, M.; Hershkowitz, R.; Ghezzi, F.; Cohen, J.; Chaim, W.; Wiznitzer, A.; Levy, J.; Leiberman, J.R.; Glezerman, M. Prolactin concentrations in preterm and term pregnancy and labour. Arch Gynecol Obstet 1996, 258, 69-74. doi: 10.1007/BF00626026.
9.) Please explain the statement, “Pasteurized donor milk had a significantly lower FSH:LH ratio compared to the preterm (p=0.003), term (p=0.005), and unpasteurized donor milk (p=0.025) (Fig. 1).
The following was added to the Discussion (Lines 429-432): “Although the Holder pasteurization did not significantly affect the levels of FSH and LH in breast milk, the procedure lowered the FSH:LH ratio compared to that found in preterm (p=0.003), term (p=0.005), and unpasteurized donor milk (p=0.025) (Fig. 1).”
10.) Using the data from Table 3: For raw milk I calculate the FSH:LH ratio as 138.69/67.01 = 2.06
For HoP milk I calculate the FSH:LH ratio as 133.39/55.54 = 2.40 which is not lower.
We calculated the ratio values dividing the measured concentration values for each sample and then calculating the mean of all ratio values. Dividing in this way produced the presented differences.
11.) Discussion: Apart from the more minor comments, above, I think the Discussion needs a major rewrite to provide clarity for the reader.
Based on your comment, the coauthors critically reviewed the Discussion, and after consultation the Discussion has been revised.
12.) Reference 9 reviews maternal diet and breast milk composition with respect to total energy, total protein, fat, fatty acids, carbohydrates, vitamins and minerals, but makes no reference to, “immunoglobulins, growth factors, cytokines, and immune cells”
13.) The same applies to reference 10.
We thank the suggestions, now Ref#1 is cited, and we have changed the Refs. #9 and #10, and the following were added: Cabinian, A.; Sinsimer, D.; Tang, M.; Zumba, O.; Mehta, H.; Toma, A.; Sant'Angelo, D.; Laouar, Y.; Laouar, A. Transfer of maternal immune cells by breastfeeding: maternal cytotoxic T lymphocytes present in breast milk localize in the Peyer's patches of the nursed infant. PLoS One 2016, 11, 0156762. doi: 10.1371/journal.pone.0156762.
Gregory, K.E.; Walker, W.A. Immunologic factors in human milk and disease prevention in the preterm infant. Curr Pediatr Rep 2013, 1, 10.1007/s40124-013-0028-2. doi: 10.1007/s40124-013-0028-2.
14.) Lines 192-193 “Prolactin is the major driver of development during pregnancy via regulation of ovarian progesterone production (in many species) and direct effects on mammary epithelial cells (in all species)” is a direct quote from the abstract of reference 13. However, it is referring to mammary gland development and not fetal or newborn development.
Thank you for this comment. After consideration, we cancelled the sentence.
15.) In the Discussion it would be preferable to highlight the implications of the results of the current study rather than the in depth description of the many findings from the literature with respect to actions of the hormones of interest on fetal and neonatal development.
Thank you for this point, we aimed to give a detailed description about the effects of different hormones, and previous observations of different studies to highlight their importance in early development. After restructuring the Discussion, the following was added: “In the present study we detected the presence of pituitary hormones, FSH, LH, and PRL throughout the first six months of lactation in breast milk produced for term or preterm infants. These hormones were measured in donor breast milk, and commercial infant formulas as well, to provide previously not available information about the FSH, LH and PRL concentration of feeding options during postnatal life. Preterm breast milk contained significantly higher amount of PRL, while FSH and LH concentrations were measured in the same amount in preterm and term breast milk. Moreover Holder pasteurization applied to ensure microbiological safety of donor milk had a significant impact on PRL concentration. Comparing different feeding options formula contained undetectable amount of pituitary hormones, donor milk contained lower PRL, than own mother’s breast milk. It is important to note, that the sensitivity of laboratory methods are constantly developing, therefore measurements analyzing the bioactive compounds of breast milk are necessary to provide detailed information about the postnatal influences impacting the newborn, and also help to supplement donor milk or develop infant formulas from new aspects. Our results, due to the detected differences between the hormonal intake of feeding forms, as it was earlier concluded (Lucas et al, 1990) strengthen the possibility that endocrine replacement might be a building block of nutritional strategies in neonatal care. An elegant study highlighted in the nineties that breastfed preterm infants achieved better results on later IQ tests, in the background numerous factors are presumable, including hormonal content of maternal milk.”
Lucas, A.; Baker, B.A.; Cole, T.J. Plasma prolactin and clinical outcome in preterm infants. Arch Dis Child 1990, 65, 977-983. doi: 10.1136/adc.65.9.977.
Lucas, A.; Morley, R.; Cole, T.J.; Lister, G.; Leeson-Payne, C. Breast milk and subsequent intelligence quotient in children born preterm. Lancet 1992, 339, 261-264. doi: 10.1016/0140-6736(92)91329-7.
Thank you for giving us this opportunity to revise and improve the manuscript based on the comments and suggestions.
Sincerely yours,
Réka A. Vass M.D. Ph.D.
